# Networks are Slacking Off: Understanding Generalization Problem in Image Deraining

## Abstract

Deep low-level networks are successful in laboratory benchmarks, but still suffer from severe generalization problems in real-world applications, especially for the deraining task. An "acknowledgement" of deep learning drives us to use the training data with higher complexity, expecting the network to learn richer knowledge to overcome generalization problems. Through extensive systematic experiments, we show that this approach fails to improve their generalization ability but instead makes the networks overfit to degradations even more. Our experiments establish that it is capable of training a deraining network with better generalization by reducing the training data complexity. Because the networks are slacking off during training, i.e. learn the less complex element in the image content and degradation to reduce the training loss. When the background image is less complex than the rain streak, the network will focus on the reconstruction of the background without overfitting the rain patterns, thus achieving a good generalization effect. Our research demonstrates excellent application potential and provides an indispensable perspective and research methodology for understanding the generalization problem of low-level vision.

## 1 Introduction

The whirlwind of progress in deep learning has produced a steady stream of promising low-level vision networks, which significantly outperform traditional methods in existing benchmark datasets. However, the intrinsic overfitting issue has prevented these deep models from real-world applications, especially when the degradation differs a lot from the training data. We call this dilemma the generalization problem. Although important, this problem is not well studied in low-level vision literature. We need more in-depth analysis and understanding, before proposing effective solutions.

Understanding generalization in low-level vision is by no means easy. It is not a naive extension of the generalization research in high-level vision. We need dedicated analysis tools to interpret new phenomena. In this paper, we hope to build a stepping stone towards a more in-depth understanding of this problem. To achieve this goal, we select a representative low-level vision task as the breakthrough point, and design quantitative analysis methods for several controlling factors. The heart of our methodology is stated as follows.

**Select deraining as the representative task.** Low-level vision includes many tasks, such as image denoising and super-resolution, which have different characteristics. A general understanding of generalization across all low-level vision tasks cannot be built in a day. Thus, we choose the image deraining task as a representative. Image deraining aims to remove the undesired rain streaks in an image. There are two considerations for selecting the deraining task. First, as a typical decomposition problem, image deraining has a relatively simple degradation model (a linear superimposition model). This will facilitate our research and enable the usage of many quantitative measurements. Second, the deraining task suffers from a severe generalization problem. Existing deraining models tend to do nothing for the rain streaks that are beyond their training distribution. See Figure 1 for an example. This phenomenon is very intuitive and easy to quantify.

**Analyze from the perspective of training data.** We argue that the generalization problem is due to the network overfitting the degradation (the rain patterns in the deraining task). The main reason for this result is the inappropriate training objective. We start our analysis with the most basic and indispensable factor in constructing the training objective – training data. There has been a lot of works trying to improve real-world performance by improving the complexity of training data. This

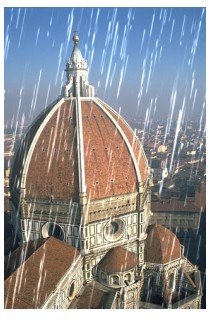 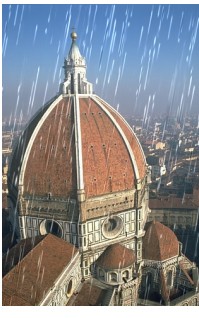 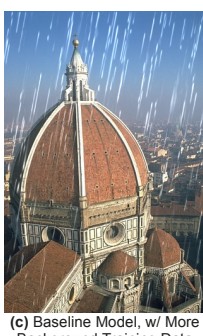 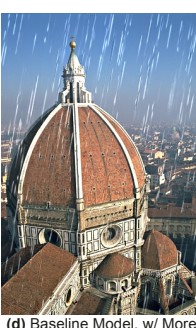 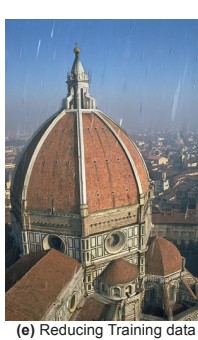

**(a)** Input Rainy Image    **(b)** Baseline Model    **(c)** Baseline Model, w/ More Background Training Data    **(d)** Baseline Model, w/ More Rain Data    **(e)** Reducing Training data According to our Conclusion

Figure 1: The existing deraining models suffer from severe generalization problems. After training with synthetic rainy images, when feeding **(a)** an image with different rain streaks, its output **(b)** shows limited effect. Two intuitive ways to improve generalization performance – **(c)** adding background images, and **(d)** adding rain patterns, cannot effectively relieve the generalization issue. In this paper, we provide a new counter-intuitive insight – **(e)** we improve the generalization ability of the deraining networks by selecting *much less* training background images for training, not more.

comes from a natural but unproven "acknowledgment" in low-level vision that more training data can solve the generalization problem. This acknowledgment also influences the deraining committee: when the network sees more (both background images and rain streaks), it can generalize better to more real-world scenarios. However, the generalization problem of deraining is NOT solved in this way. These methods still do not work on rain patterns that have not yet been collected. We argue that because too much background data is provided for training, the model cannot learn to reconstruct the image content and can only overfit to the degradation. Therefore, we propose to reduce the number of training background images in our study, rather than increase it further.

**Our analysis methods.** To systematically study the changes in model behavior brought about by changing training objectives, we construct a number of training sets consisting of different background images. We first investigate the effect on the number of training set images by overfitting the model on very few (16 even 8) images. By switching between different image categories, we study the network behaviour when fitting images of different complexity levels. We study the relationship between the complexity of the background image set and the generalization performance of deraining through extensive quantitative experiments.

Except for constructing training objectives, we also perform a fine-grained analysis of the model outputs. Previous works simply use the overall image quality as the performance indicator, such as PSNR. However, the reason for the quality deterioration may be either the unsuccessful removal of rain streaks or the poor reconstruction of the image background. Thus we decouple the deraining task as rain removal and background reconstruction, which are studied separately. Since the generalization problem in the deraining task is mainly related to the removal of rain streaks, fine-grained analysis can exclude the influence of other factors.

**Our key findings.** We find that deep networks are slacking off during training. They take shortcuts in reducing the loss, resulting in poor generalization performance. This is due to the inappropriate objective we set for training. Our key finding can be summarized as:

> *Between the image content and the additive degradation, deep networks tend to learn the less complex element in the separation task.*

Specifically, in the common training data with high background complexity and low rain complexity, the network will naturally learn to identify and separate rain streaks, because they are less complex and easier to learn. But when the real situation deviates from its depiction of rain, the network tends to ignore them and gets poor generalization performance. On the contrary, when we train the model on a less complex background image set, it exhibits better generalization ability, see Figure 1 (e). The reason is that when the complexity of the training image background is smaller than that of the rain patterns, the network will also take a shortcut to reduce the loss, i.e., remember the reconstruction of the background instead of overfitting to the rain streaks. Except for the removal of rain, the performance of the model is also determined by the background reconstruction. Reducing the background complexity of the training data could inevitably produce unsatisfactory reconstruction results. However, our results show that the model trained on only 256 images can already handle

most of the image components. These counter-intuitive phenomena have never been studied and valued in the literature.

**Implication.** Given the current literature, our results are interesting and inspiring. Our results demonstrate the importance of the training objective in determining the generalization ability. An inappropriate and incomplete training objective provides an opportunity for deep networks to "slack off". Although we hope that the network can learn the rich semantics in natural images, it is often overlooked that the low-level vision system can achieve learning goals through some shortcuts. While these shortcuts lead to poor generalization performance of the model. Our results also illustrate that a model with good generalization ability should learn the distribution of the natural images themselves, rather than overfit the degradation. By simply exploiting our findings, we can make the simplest networks exhibit excellent generalization capabilities. This shows that our findings have great potential for application.

## 1.1 RELATED WORKS

This work is first related to deraining research. But we do not propose new network structures, loss functions, or datasets like most existing deraining works. Our work is aimed at the analysis and understanding of the generalization problem in the deraining task. Due to the limited space, the deraining works are reviewed in Appendix A.1. We next review previous works about interpretability and understanding of generalization in low-level vision.

Deep learning interpretability research aims to understand the mechanism of deep learning methods and to obtain clues about the success or failure of these methods. Without a deep understanding of these working mechanisms, we are not convinced to move forward in the right direction. The research on deep learning interpretability follows a long line of works, most of them focusing on the classification task Simonyan et al. (2013); Springenberg et al. (2014); Shrikumar et al. (2017); Sundararajan et al. (2017); Zhou et al. (2018); Lundberg & Lee (2017). Most low-level vision tasks have also embraced great success with powerful deep learning techniques. There are also works on interpretability for these deep low-level networks Gu & Dong (2021); Xie et al. (2021); Magid et al. (2022). For the generalization problem in low-level vision, these problems often arise when the testing degradation model does not match the degradation used in training, e.g., different downsampling kernel Gu et al. (2019); Liu et al. (2022a); Kong et al. (2022) and noise distribution Guo et al. (2019). The existing works either develop blind restoration methods to include more degradation possibilities in the training process or make the training data closer to real-world applications. Only a little work has been proposed to study the reasons for this lack of generalization performance Liu et al. (2021; 2022b). More details of these previous works can also be found in Appendix. No research has attempted to investigate the interpretation of the training process of low-level vision networks, especially from the perspective of the generalization problem. Our work fills this gap.

## 2 ANALYSIS METHOD

Our goal is to explore how different training objectives affect the network behavior and generalization performance. Before introducing our observations, we need to describe the experimental designs and quantitative analytical methods in this section.

### 2.1 CONSTRUCTION OF TRAINING OBJECTIVE

The training objective of a deep network is jointly determined by the training data and the loss function. We set a variety of training objectives in order to observe the changes in the generalization performance of different deraining models. As shown in Figure 3 (left), a rainy image $O$ can be roughly modeled using a linear model $O = B + R$, where $B$ is the image background, and $R$ is the additive rain streaks. We will change the training objectives with different background images and rain streaks.

**Background Images.** Typically, image backgrounds are sampled from street view images Geiger et al. (2012) or natural image datasets Schaefer & Stich (2003); Arbelaez et al. (2010), as these images are close to the application scenarios of deraining. In the literature, the previous works Fu et al. (2017b); Zhang & Patel (2018) claim that the model can learn the prior knowledge of reconstructing these scenes by training on a large number of such background images. We break this common sense by constructing different training background image sets from the following two aspects.

For the first aspect, we change the number of background images. We argue that as too much background data are provided for training, the model cannot faithfully learn to reconstruct the image

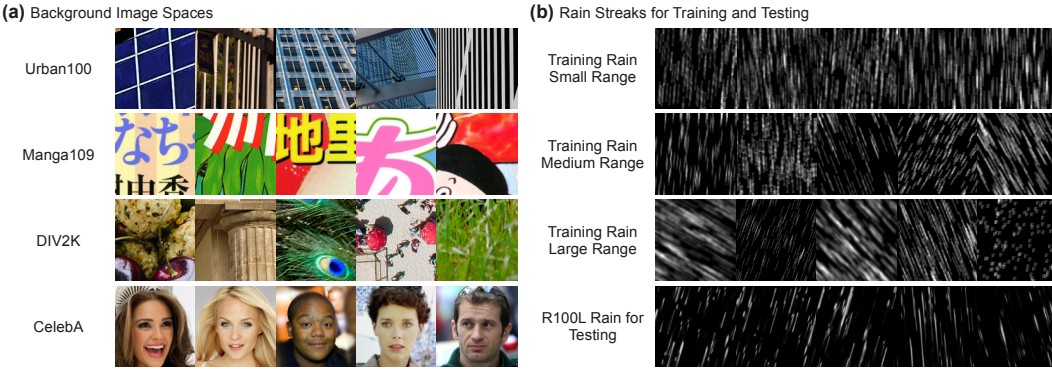

Figure 2: **(a)** Background images from different image datasets. It can be seen that the structure of the face image (CelebA) is relatively complex. Natural image patches (DIV2K) contain natural textures and patterns. The patterns in Manga109 and Urban100 are artificially created – Manga images have sharp edges, while Urban images contains a lot of repeating patterns and self-similarities. **(b)** Rain streaks used our experiments.

content but overfit the degradation patterns. To demonstrate this comment, we reduce the complexity of the background images to see how the network behavior changes in extreme scenarios. In our experiments, we use 8, 16, 32, 64, 128, 256, 512, and 1024 background image patches of size $128 \times 128$ to build the training datasets, respectively. We also use a large number of patches (up to 30,000) to simulate the common situation when the image background is sufficiently sampled.

In addition to the number of images, the image content will also affect the learning procedure. For images with many self-similar or regular patterns, it is easier for the network to remember and extract semantic features. While a face image that contains both short- and long-term dependent structures is apparently more complex than a skyscraper that consists of just repeated lines and grids Bagrov et al. (2020). We carefully choose the image distribution as the second aspect of our dataset construction. We sample from four image datasets that are distinct from each other: CelebA (face images) Liu et al. (2015), DIV2K (natural images) Timofte et al. (2017), Manga109 (comic images) Matsui et al. (2017), and Uerban100 (building image) Huang et al. (2015). Some examples of these images are shown in Figure 2 (a).

**Rain streaks synthesis.** Since it is hard to collect a large number of real-world rainy/clean image pairs, we follow the previous deraining works Garg & Nayar (2006); Fu et al. (2017b) to synthesize rainy images for research. We use

Table 1: Different rain streaks synthesis ranges.

| Range | Quantity | Width | Length | Direction |
|---|---|---|---|---|
| Small | $[200, 300]$ | $\{5\}$ | $[30, 31]$ | $[-5°, 5°]$ |
| Medium | $[200, 300]$ | $\{5, 7, 9\}$ | $[20, 40]$ | $[-30°, 30°]$ |
| Large | $[200, 300]$ | $\{1, 3, 5, 7, 9\}$ | $[5, 60]$ | $[-70°, 70°]$ |

two different kinds of rain streaks for training and testing, separately. For training, we use the computational model [1] to render the streaks left on the image by raindrops of varying sizes, densities, falling speeds, and directions. This model allows us to sample rain streaks from different distributions. We adopt three rain image ranges for training, where different ranges may lead to different generalization effects, see Figure 2 (b) for a convenient visualization and Table 1 for the detailed set-up. For testing, we use the synthetic rain patterns presented by Yang et al. (2017). Although in both cases the rain streaks are simulated and visually similar to humans, they still pose a huge generalization challenge to existing deep models.

**Loss Function.** In low-level vision, the loss function is usually defined by the difference between the output image and the ground truth. In our study, we use the $l_1$-norm loss to simplify the problem. In this setting, our conclusions do not lose generality and can be extended to other similarity-based training loss functions.

## 2.2 FINE-GRAINED ANALYSIS.

Generally, the evaluation of a deraining model is to compute similarity metrics (e.g., PSNR) between the output and ground truth images Gu et al. (2020). However, such an evaluation on the whole image may lead to unfair comparison. For example, an image with perfect background recon-

---

[1] The reimplementation of the PhotoShop rain streaks synthesis method. Please refer to https://www.photoshopessentials.com/photo-effects/photoshop-weather-effects-rain/.

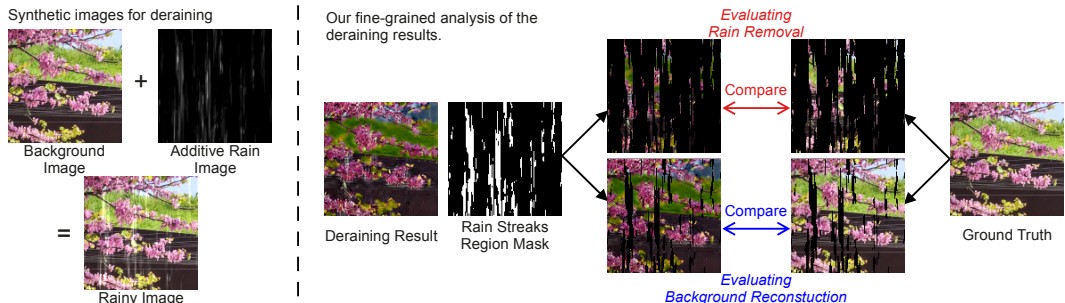

Figure 3: **(Left)** The illustration of the rainy image synthesis. **(Right)** Our fine-grained analysis of the deraining results.

struction but inferior rain removal may have a higher PSNR value than that with perfect rain removal but inferior background reconstruction (e.g., color shift). Such quantitative results would introduce systematic errors in our study.

We discuss the removal of rain streaks separately from the reconstruction of the background regions. Generalization performance of a deraining model is mainly shown in the form of removing unseen rain. The reconstruction of the background may affect the visual effect but is irrelevant to the removal of rain marks. Pixels in $R$ without rain streaks should be black, while rain streaks will appear brighter, as shown in Figure 2 (b). After synthesis, these black areas reflect the background area, while the brighter areas indicate the rainy regions. A perfect rain removal effect should do minimal damage to the background area and remove the additive signal from the rain streaks area. By processing the $R$ to a binary mask $M$ using a threshold $t$, where $M_{[i,j]} = 0$ if $R_{[i,j]} \leq t$ and $M_{[i,j]} = 1$ if $R_{[i,j]} > t$, we can segment the output image $\tilde{O}$ into the rain streaks part $\tilde{O} \odot M$ and the background part $\tilde{O} \odot (1 - M)$. We then have two performance numbers:

- *Rain Removal Performance*: $E_R = \sqrt{\mathbb{E}[(\tilde{O} \odot M - O \odot M)^2]}$ gives the effect of rain removal. A network with poor generalization will not remove rain streaks and make minimal changes to the image. This term measures the changes made by the network in the rainy regions. A higher value reflects better rain removal performance.

- *Background Reconstruction*: $E_B = \sqrt{\mathbb{E}[(\tilde{O} \odot (1 - M) - B \odot (1 - M))^2]}$ gives the effect of background reconstruction by comparing the background regions to the ground truth. A high error in this term means poor overall reconstruction quality.

## 2.3 DEEP MODELS

We summarize existing networks into three main categories. The first category is a network composed of convolutional layers and deep residual connections, and we use the ResNet Ledig et al. (2017) as a representative. The second category is the network with an encoder-decoder design, and we use UNet Ronneberger et al. (2015) as a representative. Compared with ResNet, UNet introduces down-sampling and up-sampling layers to extract global and multi-scale features, which have been proven successful in many deraining networks. The last category is image processing Transformer. Transformer Shi et al. (2022); Chen et al. (2021) is a new network structure characterized by self-attention operations. We include SwinIR Liang et al. (2021) as a representative Transformer in our study. For the training settings of these models, please check Appendix E.2.

## 3 UNDERSTANDING GENERALIZATION

In this section, we conduct experiments based on the above analysis methods. Our analysis consists of two aspects – the rain removal effect and the effect of background reconstruction.

### 3.1 GENERALIZATION ON RAIN REMOVAL

We analyze the rain removal effect on unseen rain streaks. Notably, as we use different kinds of rain streaks for training and testing, the results in this section all represent generalization performance. After comprehensive experiments, we obtain the following interesting observations.

**Training with fewer background images leads to better deraining performance.** First, we fix the range of rain streaks to medium level and then replace the background images to build different training objectives. We conduct experiments on all four categories of images. For each category,

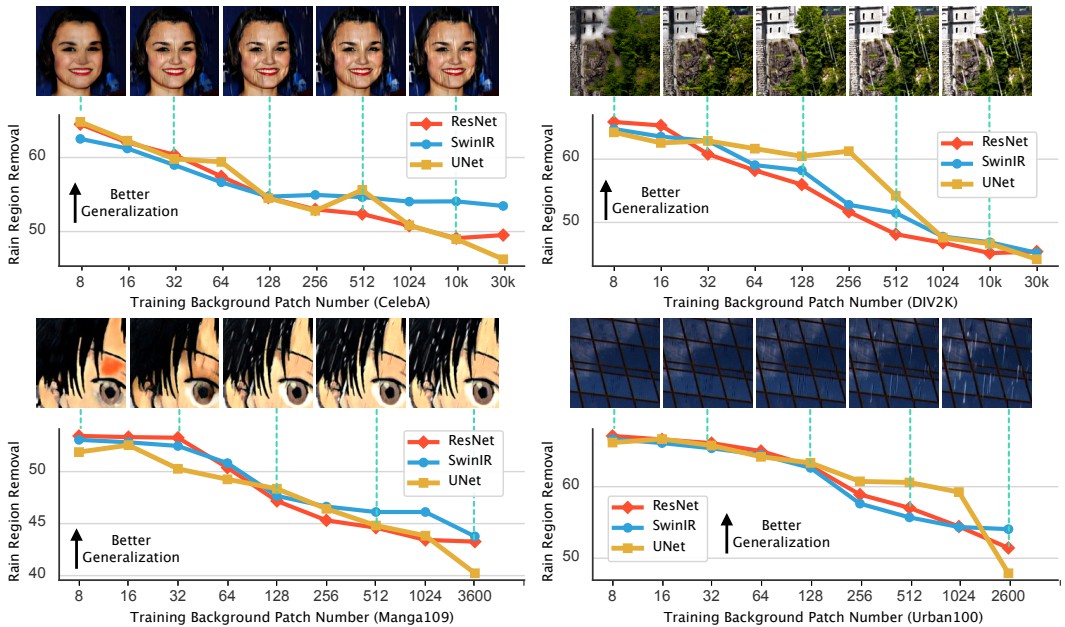

Figure 4: The relationship between the number of training patches and their rain removal performance. For each plot, the $x$-axis represents the patch number, and the $y$-axis represents the quantitative rain removal effect $E_R$. Higher values on the $y$-axis mean better rain removal. The test rain patterns are not in the training set. The effect of rain removal at this time reflects the generalization performance. The qualitative results are obtained using ResNet.

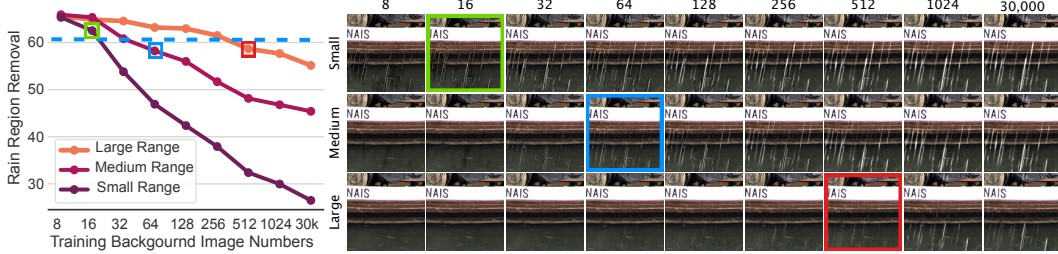

Figure 5: When trained with different rain ranges, the model exhibits different rain removal effects. The $y$-axis represents the quantitative rain removal effect. When the rain removal performance is lowered to the blue dashed line, the qualitative effect of removing rain starts to decrease significantly. We mark these cases and their corresponding effects with colors. We use ResNet in this experiment.

we use the training set with different amounts of image patches. We test the rain removal effect of these models. The testing images adopt rain streaks as in Yang et al. (2017). Their background images are sampled from each category and are different from the training set. The experimental results are shown in Figure 4. It can be seen that these experimental results speak to the same trend, despite different background images and networks. Specifically, the deraining models trained on eight image patches can surprisingly handle unseen rain streaks. On the contrary, models trained with a large number of background images cannot remove these rain streaks. This is no longer in line with our common sense. Between these two extreme states, the rain removal effect is getting worse with the increase in the number of training images. When the patch number increases to 256, the networks have already lost most of their rain removal ability. While the number of patches increased from 1024 to 30k, the rain removal effects do not change significantly (they all fail to remove rain). This can also be observed from the qualitative results.

We present our explanation of this interesting phenomenon. Although we describe the training objective as removing rain streaks from images, there are two strategies for the network to reduce the training loss. One is to recognize and remove rain streaks, and the other is to recognize and reconstruct the image background. If we do not specify the learning strategy, the network will choose the simpler one of these two strategies. When a large number of background patches are used in training, learning to reconstruct backgrounds is much more complicated than learning to remove

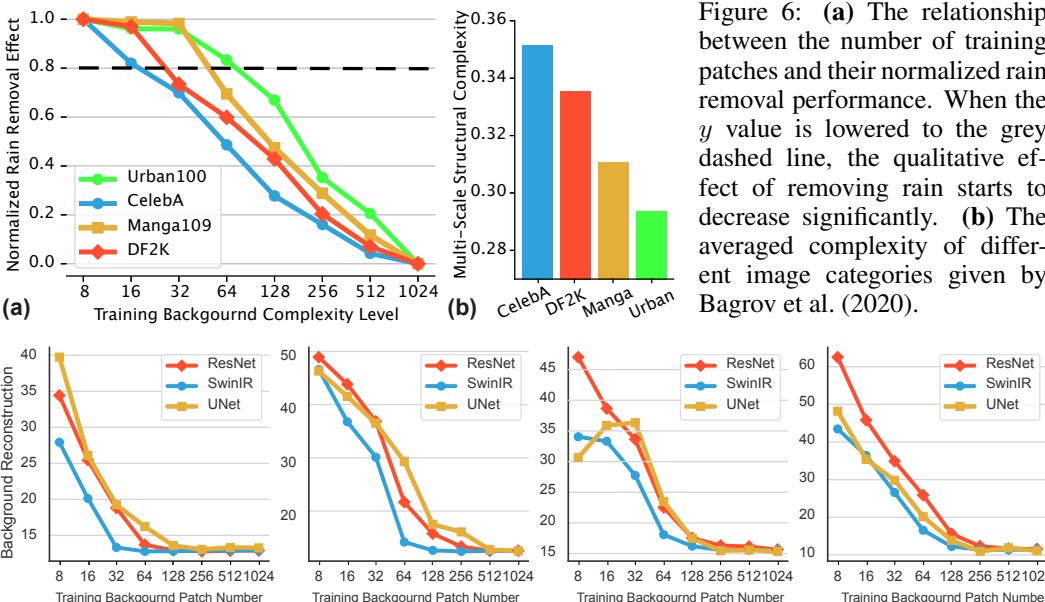

Figure 6: **(a)** The relationship between the number of training patches and their normalized rain removal performance. When the $y$ value is lowered to the grey dashed line, the qualitative effect of removing rain starts to decrease significantly. **(b)** The averaged complexity of different image categories given by Bagrov et al. (2020).

Figure 7: The relationship between the number of training patches and their background reconstruction effect. For each plot, the $x$-axis represents the patch number, and the $y$-axis represents the reconstruction error of the background.

rain. Then the network chooses to recognize rain and remove it. This will result in an overfitting problem: when new rain streaks deviate from the training ones, the network fails to recognize and remove them. On the contrary, when the background image consists of only a few image patches, learning the background is easier than learning rain streaks. The network will recognize image components in the background, and ignore the features of rain streaks. Thus, the model shows better rain removal effects in images with unseen rain streaks.

**The relative complexity between the background and rain determines the network behavior.** To verify the above conjecture, we change the range of the rain streaks used in training according to Section 2.1. When using the medium rain range, the rain removal effect weakens when training using 64 background patches. According to our explanation, a larger rain streak range makes it harder for the network to learn the rain pattern. Therefore, the rain removal effect will not be weakened until more background patches are used for training. The results of this experiment are shown in Figure 5. As can be seen, in all three training rain ranges, the rain removal effects decrease as the number of background patches increases. When there are enough background images for training (30k patches), even the large rain range of training rain cannot make the final model achieve sufficient rain removal performance, indicating that the large rain range does not cover our testing cases. When training with a large rain range, the network shows a significant drop in rain removal performance until it is trained with more than 512 background patches. The model trained on a small rain range cannot show a good rain removal effect even when there are only 16 background training image patches. These results indicate that the network behaviors are affected by the relative relationship between the background image and rain streaks. The complexity or learning difficulty of the medium range rain is approximately less than 64 training patches, while the complexity of the large range rain is approximately less than 512 training background patches. The network will slack off, and "select" the easier way to learn in different situations.

**A more complex background set makes it harder for the network to learn.** We next change the category of the background images used for training and observe the models' behavior. To compare across different image categories, we normalize the deraining effect to $[0, 1]$. The results are shown in Figure 6 (a). The most intuitive conclusion is that even with the same number of training patches, different image categories can lead to different rain removal effects. For CelebA images, when the patch number increases from 8 to 16, its deraining performance begins to drop sharply. For the natural image patches, increasing the patch number to 16 does not cause such a rain removal performance drop. In contrast, for image patches from Manga109 and Urban100, the rain removal performance does not drop significantly until the patch number is larger than 32. According to our

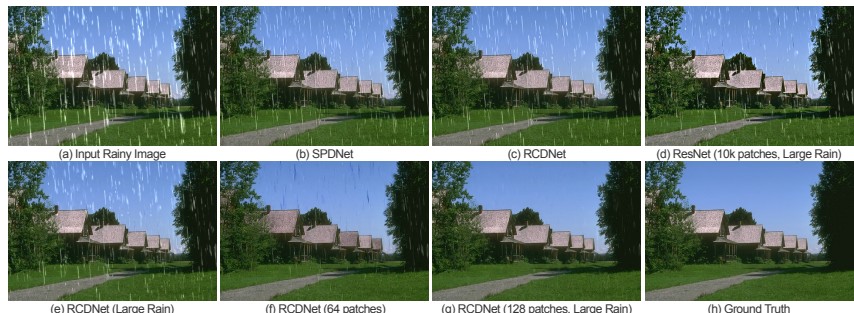

(a) Input Rainy Image     (b) SPDNet     (c) RCDNet     (d) ResNet (10k patches, Large Rain)

(e) RCDNet (Large Rain)     (f) RCDNet (64 patches)     (g) RCDNet (128 patches, Large Rain)     (h) Ground Truth

Figure 8: Visualization of the deraining results on a synthetic image. Zoom in for better comparison.

| Training | Objective | ResNet | | | SPDNet Yi et al. (2021) | | | RCDNet Wang et al. (2020c) | | |
|---|---|---|---|---|---|---|---|---|---|---|
| Back. | Range | $E_R \uparrow$ | $E_B \downarrow$ | PSNR $\uparrow$ | $E_R \uparrow$ | $E_B \downarrow$ | PSNR $\uparrow$ | $E_R \uparrow$ | $E_B \downarrow$ | PSNR $\uparrow$ |
| 30k | Medium | 31.24 | 10.79 | 25.15 | 33.63 | 5.49 | 30.51 | 26.55 | 5.41 | 28.54 |
| 64 | Medium | 53.33 | 25.02 | 20.87 | – | – | – | 45.47 | 14.78 | 25.32 |
| 512 | Large | – | – | – | 39.88 | 8.91 | 28.57 | 37.53 | 7.16 | 29.60 |
| 256 | Large | 45.64 | 16.51 | 24.30 | 38.87 | 8.03 | 29.40 | 40.40 | 8.52 | 29.08 |
| 128 | Large | 51.75 | 23.53 | 21.45 | 43.20 | 14.59 | 25.67 | 44.67 | 13.72 | 26.09 |

Table 2: Quantitative comparisons between different models. ↑ means the higher the better while ↓ means the lower the better.

explanation, as the number of training patches increases, the more complex image categories will make the models suffer from performance drop earlier. Our results suggest that the complexity of these four image categories can be in decent order as CelebA, DIV2K, Manga109 and Urban100.

This roughly matches our human perception. Face images have strong global and local structures. DIV2K images are rich in texture but have a simple global structure. Manga images lack complex textures but often contain text and complex edges. Urban images consist of repetitive stripes and grids. We then verify our conclusion with a system complexity calculated by a mathematical model. Bagrov et al. (2020) propose a computational method to estimate the structural complexity of natural patterns including natural images. We calculate the multi-scale structure complexity for these four image categories, and the results exhibit the same order, see Figure 6 (b). This provides mathematical evidence for our argument.

## 3.2 RECONSTRUCTION ON BACKGROUND

The above results show that the deraining ability can be improved by reducing the background image used for training. But training with only limited background images is not at zero cost. Reducing the training images prevents the network from overfitting the rain patterns, but causes the network to overfit the limited background images. We also conduct experiments on this issue.

With the decoupled evaluation metrics $E_B$ described in Section 2.2, we can measure the reconstruction of the background independently. The results are shown in Figure 7. With the increase of training images, the reconstruction of the background becomes better. It can be seen that training with 256 background patches can already bring a good background reconstruction effect. Continuing to add training images does not further improve the performance of background reconstruction. This conclusion is surprising, as this goes against our intuition that training the low-level vision model requires a lot of images. Our research shows that training with a large number of background images does not lead to a large gain in reconstruction performance, but instead exacerbates the model's overfitting to rain streaks. Another surprising finding is that the model trained with only 256 images can already handle most of the image components. This may indicate that image components and features are not as complex as we usually thought for a low-level vision network.

## 4 IMPLICATION

**Improve the existing deraining models.** Although this paper does not directly propose any algorithms, our conclusions can shed light on improving the removal of unknown rain streaks by existing models. Our experiments have three important practical findings: (1) Reducing the number of background images can make the network focus on learning image content instead of overfitting rain streaks. (2) Expanding the range of rain streaks allows us to use more background images for training. (3) A few background images can already achieve good reconstruction performance. These findings can be directly used to improve the generalization ability of existing models with minimal changes. Our strategy is simple: *find a balance between background images and rain range to avoid overfitting to rain streaks*.

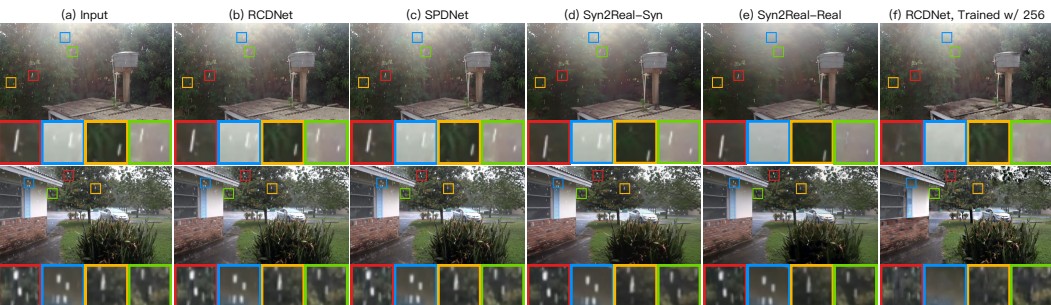

Figure 9: Qualitative results on real-world test images. Zoom in for better comparison.

Some quantitative results are presented in Table 2. We use three deraining models as baselines (ResNet, SPDNet Yi et al. (2021), RCDNet Wang et al. (2020c)) and demonstrate the power of the proposed simple strategy. We use 30K background images and the medium range rain to train our baseline models. The test set is the R100 dataset Yang et al. (2017). We quantify the deraining effect and the background reconstruction effect according to the decouple evaluation metrics $E_R$ and $E_B$. We also test PSNR as a reference. It can be seen that using the existing training methods cannot generalize well to the unseen rain of R100, which is shown by the poor deraining performance in Table 2. However, due to the learning on a large number of images, the reconstruction errors of the baseline models are generally low. Thus the PSNR values cannot objectively reflect the rain removal effect. We reduce the training background images to 64, which is the upper limit of the image number that can make the model generalize under medium range rain. At this time, the rain removal performance has greatly improved, but at the cost of background reconstruction performance. By enlarging the rain range and training with more background images, we are able to achieve a trade-off between rain removal performance and background reconstruction.

Figure 8 shows a qualitative comparison of these models under different training objectives. It can be seen that even with the advanced network structure design, the rain removal effects of the baseline models of SPDNet and RCDNet are not satisfactory. Using a larger range of rain can bring limited improvements. In the case of medium range rain, reducing the background image to 64 significantly improved the rain removal effect and resulted in unstable image reconstruction. When the rain range is enlarged, and the training background is set to 128 patches, the model can show excellent performance in rain removal and background reconstruction. Note that we do not use additional data or improve the network structure throughout the process. We only adjust the training data.

We also present the comparison on real images in Figure 9. In addition, semi-supervised methods Wei et al. (2019a); Huang et al. (2021) have also been used to improve the deraining effect on real images, and we also include the representative method Syn2Real Yasarla et al. (2020; 2021). Syn2Real-Syn is trained on synthetic data, and Syn2Real-Real is trained on synthetic labeled data and real unlabeled data. Due to the difference in the distribution of rain streaks, the models trained using synthetic data can not generate satisfactory rain removal effects. When obtaining some real images, Syn2Real-Real can indeed achieve some improvement. However, these improvements are not brought about by improving the generalization ability. Because these methods manage to convert "rain outside the training set" to "rain inside the training set". Since data collection is extremely difficult, this method still faces great challenges in practice. Our method improves generalization performance and achieves better results on test images.

## 5 CONCLUSION AND INSIGHTS

In this work, we study the generalization problem of deraining networks. Although we take image deraining as a representative, our key conclusions can provide insights for low-level vision. We argue that the generalization problem in low-level vision cannot be attributed to insufficient network capacity or training data. Instead, we find that existing training strategies do not encourage generalization. The networks only learn to overfit the degradations, and thus can hardly generalize well to unseen degradations. A feasible solution is to guide the network to learn image distributions instead of degradations. But this also poses challenges when the image distributions are complex. Furthermore, due to the lack of effective interpretability tools, we cannot explore what the low-level model learns and answer why it learns in this way. This gap between ideal and reality prompts us to rethink the importance of low-level network interpretability, and this work provides an indispensable perspective on this. These insights also need to be verified in other low-level vision tasks.

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

APPENDIX

## A  OTHER RELATED WORK

### A.1  IMAGE DERAINING

Many methods have been proposed to develop state-of-the-art deraining networks. These works include deep networks designs Fu et al. (2017a); Wang et al. (2019a), residual networks Fu et al. (2017b); Liu et al. (2019), recurrent networks Ren et al. (2019); Yang & Lu (2019); Yang et al. (2019), multi-task Wang et al. (2019c); Du et al. (2020) and multi-scale designs Jiang et al. (2020); Fu et al. (2019); Yasarla & Patel (2019); Yu et al. (2019); Wei et al. (2019b); Wang et al. (2020b); Zamir et al. (2021), sparsity-based image modeling Gu et al. (2017); Zhu et al. (2017), low-rank prior Chang et al. (2017), model-driven solutions Wang et al. (2020c;d), attention mechanism Wang et al. (2020a); Chen et al. (2021); Fu et al. (2021), adversarial learning Li et al. (2019), representation learning Chen & Li (2021), semi-supervised Yasarla et al. (2020) and unsupervised learning Chen et al. (2022). Deep learning methods are data-hungry but collecting rain streaks and background image pairs are challenging. A lot of works have been proposed to synthesize rain streaks with better results. Garg & Nayar (2006) first propose a physically-based photo-realistic rendering method for synthesizing rain streaks. Zhang & Patel (2018) and Fu et al. (2017a) use Photoshop software to manually add rain effects to images to build the synthetic paired data. Due to the poor generalization performance of existing methods, models trained on synthetic images were found to be ineffective in real-world scenarios. Some works Yang et al. (2017); Zhang et al. (2019); Wang et al. (2019b) that have contributed to real collected deraining datasets. However, acquiring these datasets is still expensive and cannot solve the problem of poor generalization. There are also works that mentioned the generalization issue of the deraining models. Xiao et al. (2021) and Zhou et al. (2021) attempt to improve the generalization ability of deraining networks by accumulating knowledge from multiple synthetic rain datasets, as most existing methods can only learn the mapping on a single dataset for the deraining task. But this attempt does not allow the network to generalize beyond the training set.

In addition, semi-supervised methods Wei et al. (2019a); Huang et al. (2021) have also been used to improve the deraining effect on real images, and we also include the representative method Syn2Real Yasarla et al. (2020; 2021). There are some semi-supervised deraining methods Wei et al. (2019a); Huang et al. (2021); Yasarla et al. (2020; 2021) are proposed to improve the performance of deraining models in real-world scenarios. When obtaining some real images similar to the test images, these works can indeed achieve some improvement. However, these improvements are not brought about by improving the generalization ability. Their solution is to include real test images in the training set, even if we don't have corresponding clean images. These methods are effective when we can determine the characteristics of the test image. But this does not solve the generalization problem. Because these methods manage to convert "rain outside the training set" to "rain inside the training set". Since data collection is extremely difficult, this method still faces great challenges in practice.

### A.2  LOW-LEVEL VISION INTERPRETABILITY

We provide a detailed review of existing work on low-level visual interpretability. Gu & Dong (2021) bring the first interpretability tool for super-resolution networks. Xie et al. (2021) find the most discriminative filters for each specific degradation in a blind SR network, whose weights, positions, and connections are important for the specific function in blind SR. Magid et al. (2022) use a texture classifier to assign patches with semantic labels, in order to identify global and local sources of SR errors. Shi et al. (2022) show that Transformers can directly utilize multi-frame information from unaligned frames, and alignment methods are sometimes harmful to Transformers in video super-resolution. They use a lot of interpretability analysis methods in their work. The closest work to this paper is the deep degradation representation proposed by Liu et al. (2021). They argue that SR networks tend to overfit to degradations and show degradation "semantics" inside the network. The presence of these representations often means a decrease in generalization ability. The utilization of this knowledge can guide us to analyze and evaluate the generalization performance of SR methods Liu et al. (2022b).

## B  TRANSFERABILITY OF LIMITED TRAINING PATCHES

At the end of the main text, we propose a method to improve the generalization performance of the deraining network by reducing the number of training background image patches. However, this method will overfit the image content when the number of training background patches is very

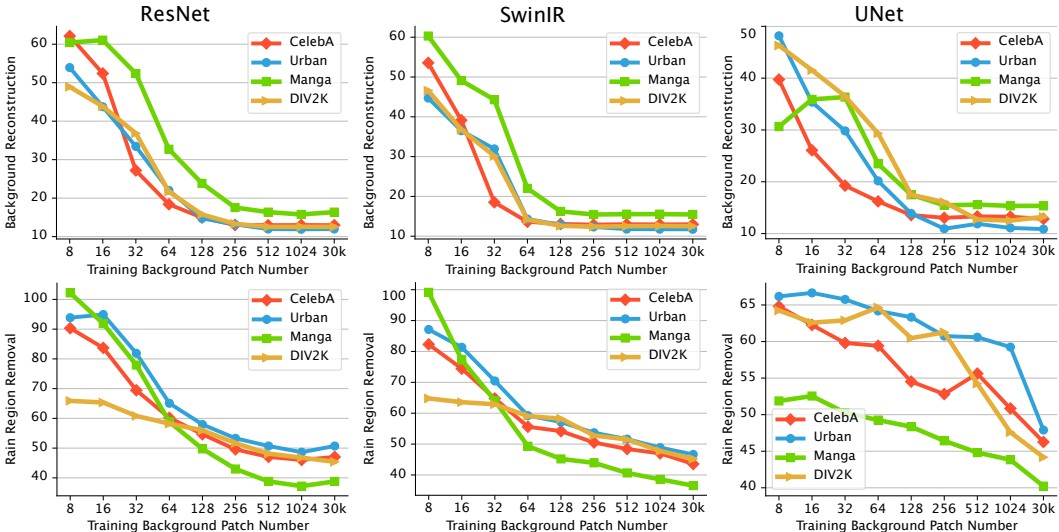

Figure 10: The relationship between the number of training patches and their rain removal or background reconstruction performance. The test image set for these six plots is the DIV2K set. We train the model with all four image categories to validate the performance when the image distribution mismatch. For background reconstruction $E_B$, lower values on the $y$-axis mean better background reconstruction. For rain removal effect $E_R$, higher values on the $y$-axis mean better rain removal. The test rain patterns are not in the training set. The effect of rain removal at this time reflects the generalization performance.

small. In Section 3.2, we investigate the risk of reducing the number of training patches when testing on the same image category. Recall that with the increase of training images, the reconstruction of the background becomes better. Training with 256 background patches can already bring a good background reconstruction effect. Continuing to add training images does not further improve the performance of background reconstruction.

In this section, we investigate whether the proposed scheme is still robust when the training and testing patch distributions are significantly different. We train the model on four image categories and then test it using the DIV2K image category. This simulates the situation when the background image distribution differs from the test set. We observe the behavior of models trained with different numbers of patches. The results are shown in Figure 10. We can draw the following conclusions. First of all, even if the distribution of training background images is very different, the model trained using the images of CelebA and Urban categories can still perform similarly to the model trained by DIV2K patches. These models can reconstruct background images well when training patches reach 128 or more. At this time, the difference in the distribution of these training sets and DIV2K does not bring significant differences. The rain removal effect of these models is also similar. Second, we found that the model trained with Manga image patches differed from others. The model trained with manga image patches is generally worse at background reconstruction than other models. Even when the number of patches is large, the model trained on manga cannot achieve similar performance to other models. For rain removal, the model trained with manga also performs the worst. This result is in line with expectations because manga images are significantly different from other images, especially in the underlying image components. Although the other three types of images differ greatly in image structure, texture type and other characteristics, they all belong to the category of natural images. Whereas Manga images contain artificial textures and edges, which are quite different from other images.

There is a large image reconstruction error when training with images whose distribution is very different from that of the test set images. This is reasonable to some extent, because in this case, even using a large number of training images cannot bridge the error caused by this distribution mismatch. And we are pleased that the method of training using limited background images is robust to image content to a considerable extent, as long as the training images are natural images. This is consistent with our practice. In the process of actually using our method, we also found that as long as more than 256 image patches are used for training, the results are stable. There is no significant performance change due to the content of the selected training patches.

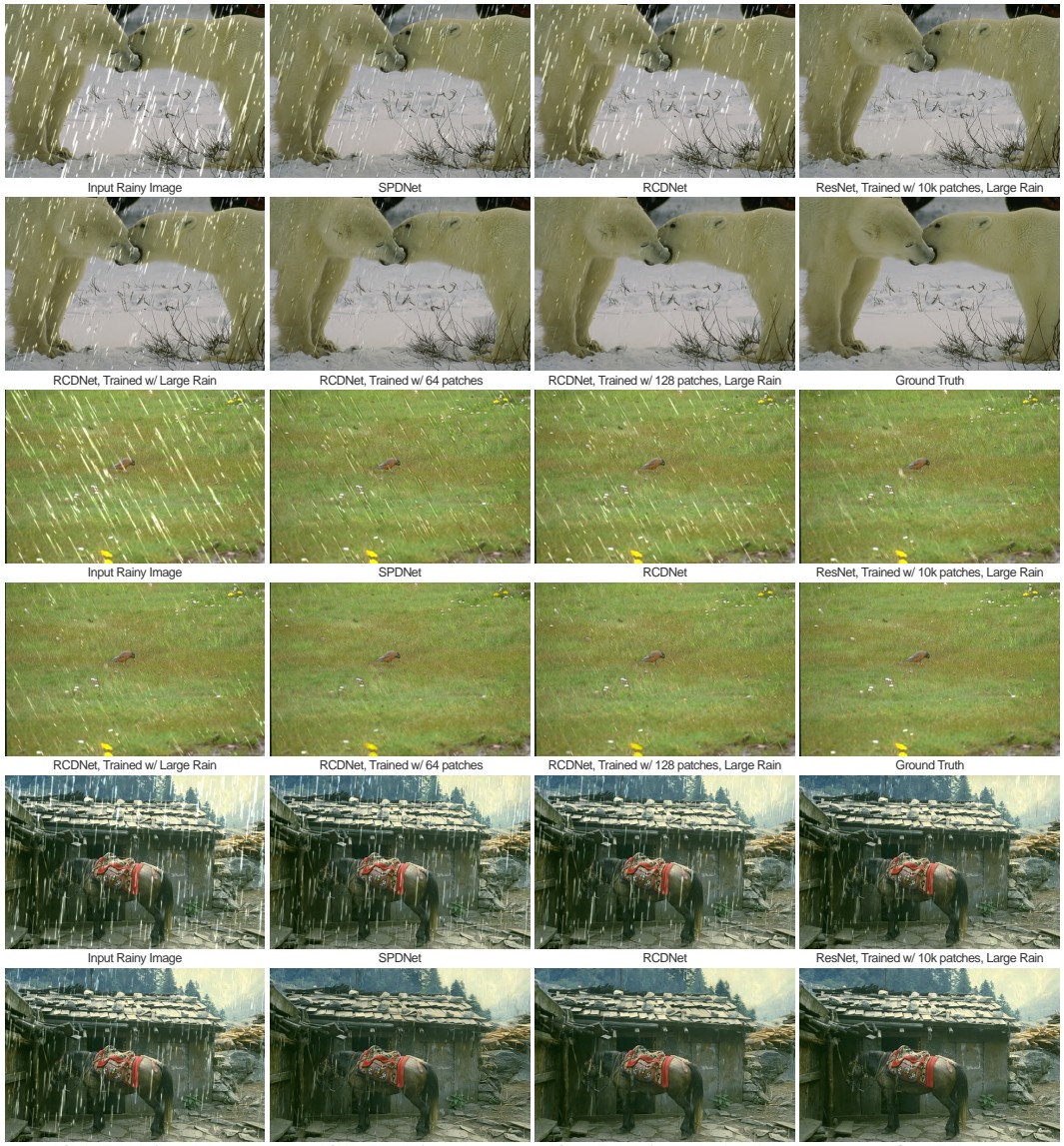

Figure 11: Visualization of the deraining results. Zoom in for better comparison.

## C    MORE RESULTS

We provide more results of different deraining models in Figure 11 and Figure 12. Note that we did not use additional data nor improve the network structure throughout the process. We only adjust the training objective. Although the effect of the output image can be further improved, it shows our conclusions' practical value and application potential.

## D    LIMITATION.

Our work mainly takes the deraining task as a breakthrough point, and attempts to make a general summary of the generalization problem in low-level vision. Due to the differences between different low-level tasks, the analysis methods in this paper, especially the fine-grained analysis methods, may not be directly used on some other tasks. But we believe our work can still bring novel insights to the entire low-level vision field.

Our work also attempts to improve existing deraining models. But these improvements are based on the simple usage of some key conclusions of our work. Although shown effective, we believe that these methods are still far from ideal. We only demonstrate the application potential of the knowledge presented in this work and have no intention to propose state-of-the-art algorithms or

models. Research efforts are still needed to develop more robust deraining algorithms using our conclusions.

# E    REPRODUCIBILITY STATEMENT

## E.1    RESOURCES

The models used in our work are taken directly from their respective official sources. Our code is built under the BasicSR framework `https://github.com/xinntao/BasicSR` for better code organization. The deraining model SPDNet Yi et al. (2021) is available at `https://github.com/Joyies/SPDNet`. The deraining model RCDNet Wang et al. (2020c) is available at `https://github.com/hongwang01/RCDNet`. The training and testing datasets used in our work are all publicly available.

## E.2    NETWORK TRAINING

Due to space constraints, we do not describe our training method in detail in the main text. Here we describe the training method to reproduce our results. A total of 150 models were involved in our experiments. We used the same training configuration for all models. We use Adam for training. The initial learning rate is $2 \times 10^{-4}$ and $\beta_1 = 0.9$, $\beta_2 = 0.99$. For each network, we fixed the number of training iterations to 250,000. The batch size is 16, input rainy images are of size 128×128. The cosine annealing learning strategy is applied to adjust the learning rate. The period of cosine is 250,000 iterations. All models are built using the PyTorch framework Paszke et al. (2017) and trained with NVIDIA A100 GPUs.

## E.3    AVAILABILITY

All the trained models and code will be publicly available.

# F    ETHICS STATEMENT

This study does not involve any human subjects, practices to data set releases, potentially harmful insights, methodologies and applications, potential conflicts of interest and sponsorship, discrimination/bias/fairness concerns, privacy and security issues, legal compliance, and research integrity issues. We do not anticipate any direct misuse of our contribution due to its theoretical nature.

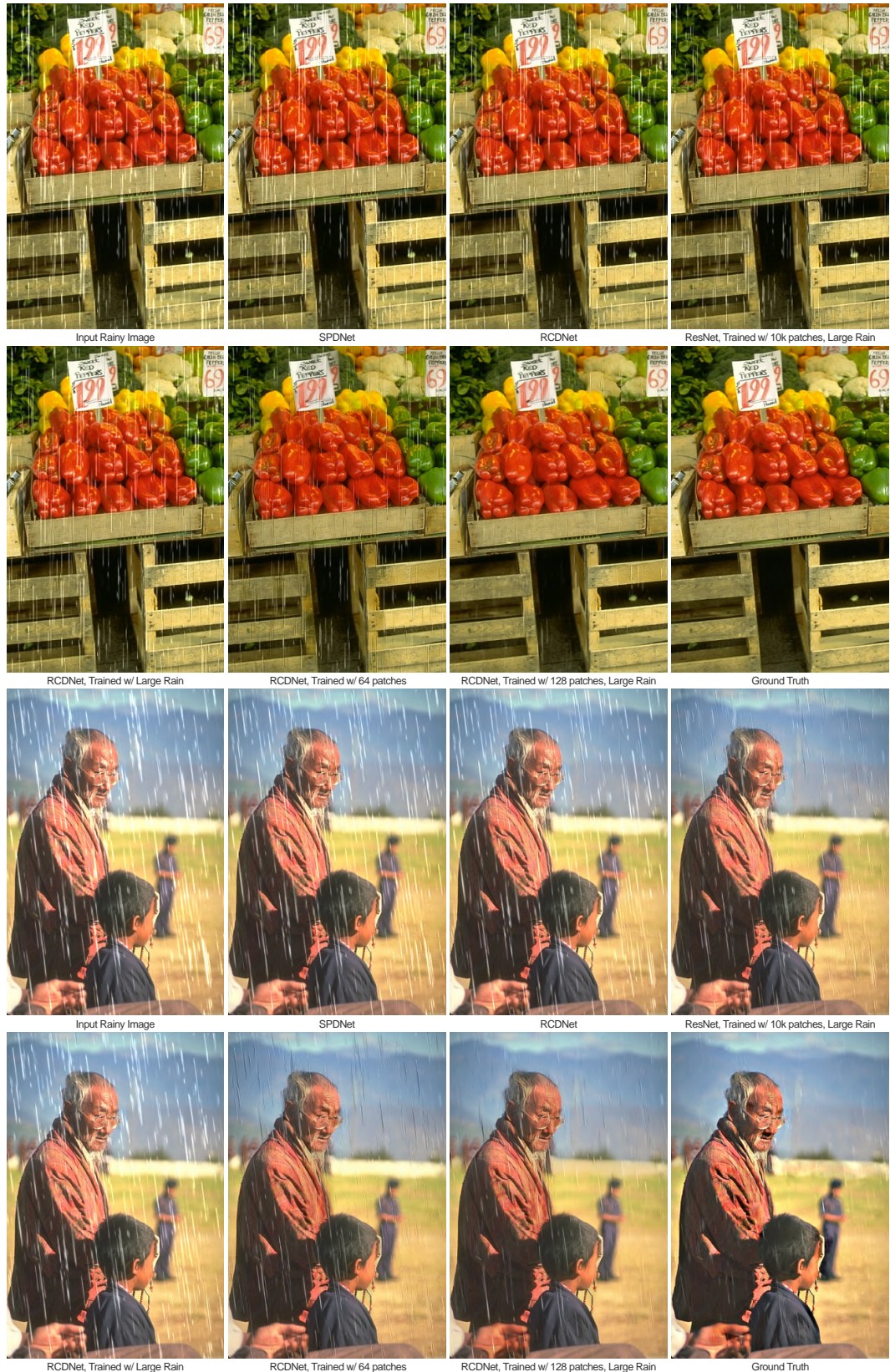

Figure 12: Visualization of the deraining results. Zoom in for better comparison.

