# OpenReview forum: "Networks are Slacking Off: Understanding Generalization Problem in Image Deraining"
_ICLR.cc/2023/Conference — Submitted to ICLR 2023_

### Official Review · Reviewer_kgGu · 2022-10-20

**Confidence:** 4
**Correctness:** 3
**Technical Novelty And Significance:** 3
**Empirical Novelty And Significance:** 3
**Recommendation:** 6

**Clarity, Quality, Novelty And Reproducibility:**

This paper is well written and easy to understand. The idea that the authors improve generalization by reducing the complexity of background images is interesting.

**Strength And Weaknesses:**

Strength：
+ The authors' perspective on solving the generalization problem is interesting.

Weaknesses:
- Lack of experiments on real-world rainy images.
- There are many typos. The author should polish the manuscript.

**Summary Of The Paper:**

This paper proposes a new solution to the generalization problem of image de-raining, that is, focusing on learning the image background  can improve the generalization ability. Based on the above conclusions, the author conducts experiments by modifying the objective function and training samples to verify the effectiveness of the proposed training scheme.

**Summary Of The Review:**

Overall, the idea so this paper is somewhat novel. However, the generalization problem in the field of image deraining lies in the difference between sythetic data and real-world data,  which has not been verified by experiments in this paper.

---

> ### Author Response · Authors · 2022-11-17
> **Response to Reviewer kgGu**
>
> We thank reviewer kgGu for his/her review time and comments. Below we provide our point-by-point responses to reviewer's detailed remarks, which we hope can address his/her current reservations:
>
> `Q1`: "Lack of experiments on real-world rainy images."
>
> `A1`: Thanks for the comments. In an updated version of the paper, we include test results on real images (Figure 9). We compare two existing methods with supervised training (RCDNet and SPDNet), and semi-supervised methods for deraining. Among them, Syn2Real-Syn is trained on complete synthetic data, and Syn2Real-Real is trained on synthetic labeled data and a part of real SIRR unlabeled data. The real images are from the SPA dataset. It can be seen that due to the difference in the distribution of rain streaks, the models trained using synthetic data can not generate a satisfactory rain removal effect. When obtaining some real images, Syn2Real-Real can indeed achieve some improvement. However, these improvements are not brought about by improving the generalization ability.
>
> `Q2`: "There are many typos. The author should polish the manuscript."
>
> `A2`: Thanks for pointing it out. We revised the whole paper to reduce typos.
>
> `Q3`: "However, the generalization problem in the field of image deraining lies in the difference between sythetic data and real-world data, which has not been verified by experiments in this paper."
>
> `A3`: Thanks for the comment but we respectfully disagree with your comment. In our work, we mainly use two experiments to demonstrate the generalization improvement of our method compared with existing methods. First, we train the model using fixed training data and test it on data other than the training set. The experimental results are shown in Figure 8. Although the test and training images are all synthesized in this experiment, many models cannot generalize due to different rain synthesizing methods. Our method here demonstrates good generalization potential. The second experiment is conducted on real test images. The experimental results are shown in Figure 9. This is exactly what the reviewer has mentioned "the difference between sythetic data and real-world data". It can be seen that our method still achieves good results. Whereas our method is only trained on synthetic data. This is also good evidence to demonstrate the potential of our method in terms of generalization performance. In contrast, Syn2real-Real, which uses a semi-supervised training method and has seen real data during training, does not show comparable generalization performance on the corresponding images. Because these methods manage to convert "rain outside the training set" to "rain inside the training set". These methods still do not solve the generalization problem. Our method improves generalization performance and achieves better results on test images. We believe these experiments have demonstrated the potential of our findings in improving generalization performance.
>
> `Additional Comment`:
>
> We notice that the reviewer thinks "Some of the paper’s claims have minor issues". We would like to ask if there are any issues other than those discussed above? If the reviewers can point it out, we can explain and discuss it further in the following discussion time. We hope to address any concerns of the reviewer.

---

> ### Author Response · Authors · 2022-11-22
> **Further discussion with Reviewer kgGu**
>
> Dear reviewer kgGu:
>
> We thank you for the precious review time and valuable comments. We have provided corresponding responses and results, which we believe have covered your concerns. We realized that we could no longer revise our manuscript. But we can continue the discussion to address your concerns.
>
> We hope to discuss further with you whether or not your concerns have been addressed. Please let us know if you still have any unclear parts of our work.
>
> Best,
>
> Paper 716 Authors.

---

### Official Review · Reviewer_LTSX · 2022-10-24

**Confidence:** 5
**Correctness:** 2
**Technical Novelty And Significance:** 3
**Empirical Novelty And Significance:** 4
**Recommendation:** 6

**Clarity, Quality, Novelty And Reproducibility:**

The finding is interesting. If it is solid, it may contribute to a wide range of low level vision tasks.

**Strength And Weaknesses:**

Strengths: without modification on network architecture, existing deraing methods can be trained to have better generalization ability by reducing background images in training data. It is interesting and surpersing.

Weakness:
1. Although it is interesting, this finding is dangous to works for specific data and task. If it is solid for deraining, this paper also contributes to this field. But the training data and testing data distribution should be well discussed and evaluated. If the distribution of background images in training and testing data, the conclusion has nothing to endorse. Also, in Fig. 7, when patch number continues to increase, reconstruction error will increase?

2. Experiments are not sufficient to convince me. Cross-evaluation should be conducted on public datasets to show generalization ability of existing deraining models are indeed improved.

3. typos: Sec 3.2 that deraining the deraining ability

**Summary Of The Paper:**

This paper has an interesting finding for improving generalization ability of image deraining. By reducing the background images in training data, the deraining model will tend to learn background context instead of learning rain streaks, preventing deraining model overfitting to rain streaks patterns. The finding is interesting and may help improve generalization ability of restoration models in other low level vision tasks.

**Summary Of The Review:**

The finding is interesting, but it needs more validations to convince that the conclusion is solid. With the current experiments, it is not sufficient to support the finding is correct for deraining task. If the validation cannot be done during rebuttal, I will not suggest to accept it, because it is dangeous to mislead the comminity.

---

> ### Author Response · Authors · 2022-11-17
> **Response to Reviewer LTSX (Part 2)**
>
> `Q3`: "Experiments are not sufficient to convince me. Cross-evaluation should be conducted on public datasets to show generalization ability of existing deraining models are indeed improved."
>
> `A3`: Thanks for the comment. In the existing deraining research, there are already some fixed evaluation methods, but these evaluation methods cannot be used to evaluate the generalization ability of the deraining model. For the existing test set, a corresponding training set is also proposed. Existing work generally publishes multiple models trained on different training sets for testing on different test sets. But this does not reflect generalization ability.
> In our work, our test set is a public test set [R1]. The training set does not correspond to this public test set. This can reflect the model generalization performance when the training set is different from the test set. Our visual effect comparison can also reflect this conclusion. We provide a comparison of performance on this test set (Figure 8). In the revised version of the paper, we also compare our method with existing methods on real images (Figure 9). This comparison is even unfair to our model, since our method has only seen an extremely limited set of training images (256 training patches) and patterns of rain streaks. While other models are trained on a large amount of data, the Syn2Real-SIRR method is even compared when the real test images are available. This further illustrates the potential and power of our simple approach.
>
> We understand and appreciate the caution of the reviewers. Indeed, conclusions without adequate validation can mislead the community. We are very confident that the fundamental findings demonstrated by our systematic experiments are sound. Our conclusions are not drawn by evaluating the performance of our final method, but by well-designed research experiments. Our method is only an extension of our conclusions. While also showing solid results, it should not be confused with our conclusions.
>
> We also want to remind the reviewer. This paper is trying to study something that is novel, something that the community needs, and something for which there may not be a definitive answer. It may take years for the community to achieve an "off-the-shelf" description or conclusion about the problem we are working on and perfectly address the reviewers' concerns. But it was too late. Currently, our results and discussions are acceptable and valuable to the community. With great humility and prudence, we also feel that this paper is not the end of the problem we are working on, but a starting point. We hope that the reviewer will consider what this paper successfully reveals, and the potential value and vitality that the findings of this paper bring to the community. There is little interpretability research on low-level vision networks. We hope this work will encourage more exploration of the field.
>
> `Q4`: "typos: Sec 3.2 that deraining the deraining ability"
>
> `A4`: Thanks for pointing it out. We have fixed it. We also revised the whole paper to reduce typos.
>
> `Additional Comment`:
>
> We notice that the reviewer thinks "Several of the paper’s claims are incorrect or not well-supported". We would like to ask if there are any issues other than those discussed above? If the reviewers can point it out, we can explain and discuss it further in the following discussion time. We hope to address any concerns of the reviewer.
>
> `Reference`:
>
> [R1] Deep joint rain detection and removal from a single image. InProceedings of the IEEE conference on computer vision and pattern recognition 2017 (pp. 1357-1366).

---

> ### Author Response · Authors · 2022-11-17
> **Response to Reviewer LTSX (Part 1)**
>
> We thank reviewer LTSX for his/her review time and comments. Below we provide our point-by-point responses to the reviewer's detailed remarks, which we hope can address your current reservations:
>
> `Q1`: "But the training data and testing data distribution should be well discussed and evaluated. If the distribution of background images in training and testing data, the conclusion has nothing to endorse."
>
> `A1`: Thanks for the comment. It seems that part of the key information in the question is missing, but we guess the reviewer should be concerned about whether our findings hold when the training and testing data distributions do not match. We conduct additional experiments for this question.
>
> At the end of the main text, we propose a method to improve the generalization performance of the deraining network by reducing the number of training background image patches. However, this method will overfit the image content when the number of training background patches is very small. Section 3.2 investigates the risk of reducing the number of training patches when testing on the same image category. Recall that the increase of training images improves the background reconstruction. Training with 256 background patches can already bring a good background reconstruction effect. Continuing to add training images does not further improve the performance of background reconstruction.
>
> We investigate whether the proposed scheme is still robust when the training and testing patch distributions are significantly different. We train the model on four image categories and then test it using the DIV2K image category. This simulates the situation when the background image distribution differs from the test set. We observe the behavior of models trained with different numbers of patches. The results are shown in Figure 10. We can draw the following conclusions. First of all, even if the distribution of training background images is very different, the model trained using the images of CelebA and Urban categories can still perform similarly to the model trained by DIV2K patches. These models can reconstruct background images well when training patches reach 128 or more. At this time, the difference in the distribution of these training sets and DIV2K does not bring significant differences. The rain removal effect of these models is also similar. Second, we found that the model trained with Manga image patches differed from others. The model trained with manga image patches is generally worse at background reconstruction than other models. Even when the number of patches is large, the model trained on manga cannot achieve similar performance to other models. For rain removal, the model trained with manga also performs the worst. This result is in line with expectations because manga images are significantly different from other images, especially in the underlying image components. Although the other three types of images differ greatly in image structure, texture type and other characteristics, they all belong to the category of natural images. Whereas Manga images contain artificial textures and edges, which are quite different from other images.
>
> There is a large image reconstruction error when training with images whose distribution is very different from that of the test set images. This is reasonable to some extent, because in this case, even using a large number of training images cannot bridge the error caused by this distribution mismatch. And we are pleased that the training method using limited background images is robust to image content to a considerable extent, as long as the training images are natural images. This is consistent with our practice. In the process of actually using our method, we also found that as long as more than 256 image patches are used for training, the results are stable. There is no significant performance change due to the content of the selected training patches.
>
> We believe this experiment can alleviate the reviewer's concerns about our method's background reconstruction. Even in Figure 9, our method achieves better image reconstruction while deraining better. These discussions have been updated to a newer version of the paper. We thank the reviewer for the comment.
>
> `Q2`: "in Fig. 7, when patch number continues to increase, reconstruction error will increase?"
>
> `A2`: In Fig. 7, on all four image categories, the error of background reconstruction decreases as the training patch number increases. In Fig. 7, we only show the case up to 1024 patches. In our experiments, we tested up to 30,000 patches. Experiments show that even using 30,000 patches does not further reduce the background reconstruction error significantly. Of course, the error will not increase. We present some results here.
>
> | Dataset | 512 patches | 1024 patches | 10,000 patches | 30,000 patches |
> |---|---|---|---|---|
> | DIV2K | 12.523 | 12.547 | 12.521 | 12.528 |
> | Face | 12.913 | 12.883 | 12.913 | 12.852 |

---

> ### Author Response · Authors · 2022-11-22
> **Further discussion with Reviewer LTSX**
>
> Dear reviewer LTSX:
>
> We thank you for the precious review time and valuable comments. We have provided corresponding responses and results, which we believe have covered your concerns. We realized that we could no longer revise our manuscript. But we can continue the discussion to address your concerns.
>
> We hope to discuss further with you whether or not your concerns have been addressed. Please let us know if you still have any unclear parts of our work.
>
> Best,
>
> Paper 716 Authors.

---

### Official Review · Reviewer_DKEu · 2022-10-25

**Confidence:** 5
**Correctness:** 3
**Technical Novelty And Significance:** 3
**Empirical Novelty And Significance:** 3
**Recommendation:** 6

**Clarity, Quality, Novelty And Reproducibility:**

The idea of this paper is novel and the finding it proposed are in great conflict with existing understanding, which has not been involved before. This paper writing is good with better organizations and presentations. The experiments are organized well and the analysis are convincing.

**Strength And Weaknesses:**

Strength ：
1.	The advantage of this paper is that it tries to improve the generalization performance of images deraining by changing the training objective, and finds that training with fewer data and less complex background images can improve the generalization in deraining.
2.	Different from the existing acknowledgement, this method doesn’t need to train the deraining networks with amounts of data and extremely complex background images and rain streaks.

Weaknesses ：
1.	The disadvantage is that although this paper finds the impact of the training objective on the performance of rain removal generalization, it doesn’t propose a better way to achieve the balance between restoring the image background and removing rain streaks.
2.	It doesn’t show a significant improvement in generalization performance compared to the existing rain removal methods.
3.	I think this paper needs to do more experiments to prove the effectiveness of this finding.


**Summary Of The Paper:**

This paper proposed a new finding that between the image content and the additive degradation, deep networks tend to learn the less complex element in the separation task. To solve the generalization problem in image deraining, it tried to train the deraining networks with the fewer and less complex background images.
In my opinion, the proposed finding is novel and effective to explore the generalization problem in images deraining, which utilizes less training data and less complex background images, and this finding differs from previous acknowledgement that deep learning methods need to use the training data with higher complexity to learn richer knowledge to overcome generalization problems. Furthermore, to prove this finding, the paper conducts extensive experiments, including carefully choose the different images’ background and design different rain streaks. The experimental design and the analysis are clear to understand. The paper writing is also professional with idea organizations and presentations.


**Summary Of The Review:**

The finding in this paper is novel and subversive, but it doesn’t propose a better way to achieve the balance between restoring the image background and removing rain streaks. It doesn’t show a significant improvement in generalization performance compared to the existing rain removal methods. As to the experiment results shown in the paper, I don’t find this method is superior to other deraining methods. I think this paper needs to do more experiments to prove the effectiveness of this finding.

---

> ### Author Response · Authors · 2022-11-17
> **Response to Reviewer DKEu (Part 2)**
>
> `Q4`: "I think this paper needs to do more experiments to prove the effectiveness of this finding."
>
> `A4`: Thanks to the reviewer for the comment. In this paper, we performed extensive experiments to verify the soundness and effectiveness of our findings. In terms of soundness, we quantitatively studied the relationship between the training background image and the rain removal effect (Figure 4), the relationship between the training rain streak ranges and the rain removal effect (Figure 5), and the relationship between the training background image complexity and the rain removal effect (Figure 6). We also quantify the relationship between training images and background reconstruction performance (Figure 7). In terms of effectiveness, we show numerical results comparing our model's deraining and background reconstruction effects and previous works (Table 2). We demonstrate the generalization ability of different models and existing works of our method on synthetic images (Figure 8). We also demonstrate the performance of our method on real images and compare it with existing methods trained with supervised training and semi-supervised training methods (Figure 9). We believe that these experiments are sufficient to support this paper's core arguments and demonstrate our findings' value and effectiveness.
>
> `Q5`: "The finding in this paper is novel and subversive, but it doesn’t propose a better way to......"
>
> `A5`: Again, we thank the reviewers for considering our findings as "novel" and "subversive". But we would like to defend the other comments made by the reviewer. In our work, we demonstrate the potential of directly implementing our findings. We can further devise more complex and elaborate methods for better results, but this is beyond the scope of this paper and also beyond the limit of a 9-page paper. Future works will apply our ideas to designing generalizable low-level visual methods. While we are always looking for better methods, we believe that interpretability work like ours is equally valuable and indispensable. Any other concerns in your summary are explained in the first few questions. Please review the response before. We thank the reviewer's effort and precious time.
>
> `Additional Comment`:
>
> We notice that the reviewer thinks "Some of the paper’s claims have minor issues". We want to ask if there are any issues other than those discussed above. If the reviewers can point it out, we can explain and discuss it further in the following discussion.

---

> ### Author Response · Authors · 2022-11-17
> **Response to Reviewer DKEu (Part 1)**
>
> We thank reviewer DKEu for his/her review time and comments. Below we provide our point-by-point responses to the reviewer's detailed remarks, which we hope can address his/her current reservations.
>
> `Q1`: "The advantage of this paper is that it tries to improve the generalization performance of images deraining by changing the training objective"
>
> `A1`: We thank the reviewer for acknowledging the contribution of this paper. But we think we need to point out that we don't claim that our work "solves the generalization problem", or that the most important purpose of our work is to "improve the generalization of the model". The primary purpose of this paper is to improve our understanding of the low-level vision generalization problem. We think this is as important as proposing a specific method. The method used in our final experiments is quite simple, as simple methods often reveal more fundamental insights. Not to mention that our method reflects good generalization performance, even compared to many existing methods. We would still like to emphasize the importance of developing more advanced methods in future works based on our findings.
>
> `Q2`: "The disadvantage is that although this paper finds the impact of the training objective on the performance of rain removal generalization, it doesn’t propose a better way to achieve the balance between restoring the image background and removing rain streaks."
>
> `A2`: Thanks for the comment. On the one hand, we describe a method according to our findings to improve the generalization performance and conduct experiments at the end of the paper. The specific method is to reduce the background images' complexity while increasing the rain pattern's complexity in the training set. We tested several approaches and found that quite good results can be obtained when using 256 training patches and the large range of training rain streaks. This holds true even for real-world testing images. (We have included comparisons on real images in the revised version of the paper, please check it out.) Our performance actually speaks to the value of our findings.
>
> On the other hand, our method is robust to the choice of rain streaks and background images. Our test method uses 256 image patches for training. But in practice, the number of background images and the content of the images can vary greatly without having a big impact on the final results. In other words, "the balance between restoring image background and removing rain marks" can be easily achieved. Simple experimentation can determine a dessert proportion.
>
> `Q3`: "It doesn’t show a significant improvement in generalization performance compared to the existing rain removal methods."
>
> `A3`: Thanks for the comment but we respectfully disagree with your comment. In our work, we mainly use two experiments to demonstrate the generalization improvement of our method compared with existing methods. First, we train the model using fixed training data and test it on data other than the training set. The experimental results are shown in Figure 8. Although the test and training images are all synthesized in this experiment, many models cannot generalize due to different rain synthesizing methods. Our method here demonstrates good generalization potential. The second experiment is conducted on real test images. The experimental results are shown in Figure 9. It can be seen that our method still achieves good results. Whereas our method is only trained on synthetic data. This is also good evidence to demonstrate the potential of our method in terms of generalization performance. In contrast, Syn2real-Real, which uses the semi-supervised training method and has seen real data during training, does not show comparable generalization performance on the corresponding real images. We believe these experiments have demonstrated the potential of our findings in improving generalization performance.

---

> ### Author Response · Authors · 2022-11-22
> **Further discussion with Reviewer DKEu**
>
> Dear reviewer DKEu:
>
> We thank you for the precious review time and valuable comments. We have provided corresponding responses and results, which we believe have covered your concerns. We realized that we could no longer revise our manuscript. But we can continue the discussion to address your concerns.
>
> We hope to discuss further with you whether or not your concerns have been addressed. Please let us know if you still have any unclear parts of our work.
>
> Best,
>
> Paper 716 Authors.

---

### Official Review · Reviewer_bcPF · 2022-10-25

**Confidence:** 5
**Correctness:** 3
**Technical Novelty And Significance:** 3
**Empirical Novelty And Significance:** 3
**Recommendation:** 5

**Clarity, Quality, Novelty And Reproducibility:**

- The experimental study is good to show the generalization ability of the network while training with different types of data.
- but conclusions and implications are naive
- Please refer to weaknesses for detailed explanation

**Strength And Weaknesses:**

Strengths:
- For the experimental authors conducted on three different types of networks, U-Net, ResNet, SWIN-IR. these experiments are beneficial in understanding the behavior of different types of networks in different training conditions.
- Compares rain regions and backgrounds with different metrics (Rain Removal Performance and Background Reconstruction). But these metrics are just l2-loss in background regions and rain regions which are not excellent metrics since they don't measure the quality of background structures or quality of rain regions, and networks are prone to degrade background structures by over deraining.
- Paper is well written with a detailed study of each experiment performed.
Weaknesses:
- section 4 Implications:
     - Observations from the experiments are not excellent: "Our strategy is simple: find a balance between background images and rain
range to avoid overfitting to rain streaks." This conclusion statement is naive since this statement is very obvious and doesn't require any experiments and study to get to this conclusion.
    - It would have been better if authors propose a training strategy or a loss that addresses generalization problems in low-level visions
    - Even though the experimental study is good, the conclusion and implications are not contributing to addressing the generalization problems of low-level vision.
- Authors claim that networks have generalization problems in addressing deraining, but few existing works are proposed different generalization techniques in overcoming these generalizations differences between synthetic rain and real rain. Can authors perform similar experiments and study these semi-supervised deraining methods? Since it is important to know for the reader whether these semi-supervised deraining methods are overcoming these generalization problems in low-level vision
- Experiments on real rainy images are not performed.
- Can authors also discuss how their study can be generalized to other low-level vision tasks like de-hazing, desnowing, low-light enhancement? Since most of the comparisons and experiments are more specific to deraining.

**Summary Of The Paper:**

The paper presents generalization problems of the network in performing the low-level vision task, i.e. authors perform a study by performing the different deraining experiments and convey that when trained with a less complex background, networks learn the less complex element in the image content and degradation to reduce the training.  The paper finally claims that authors experimental study demonstrates an excellent methodology for understanding the generalization of the problem of low-level vision.

**Summary Of The Review:**

- This is good work but can be improved before publishing the work

---

> ### Author Response · Authors · 2022-11-06
> **The First Response to Reviewer bcPF (Part 2)**
>
> `Q3`: "Even though the experimental study is good, the conclusion and implications are not contributing to addressing the generalization problems of low-level vision."
>
> `A3`: We respectfully disagree with your comments. Our findings relate to the root cause of generalization problems in low-level vision. Understanding the cause of the problem is an important first step in solving the problem. For the deraining task, we also directly apply our findings and get positive results. For some other low-level vision tasks, our core view still holds; that is, the generalization problem is caused by the unreasonable setting of the training objective, which leads to the overfitting of the deep model to the degradation. Although we did not try it directly on other tasks (other tasks may not have such an easy-to-analyze feature as the deraining task), our conclusions can still provide new thinking given the similarities between the low-level vision tasks. This is also recognized by other reviewers.
>
> `Q4`: "Can authors perform similar experiments and study these semi-supervised deraining methods?"
>
> `A4`: Thanks for the suggestion. We will discuss our work with semi-supervised deraining methods [R1, R2, R3]. As mentioned by the reviewer, these works are also proposed to improve the performance of deraining models in real-world scenarios. When obtaining some real images similar to the test images, we believe that these works can indeed achieve some improvement. However, these improvements are not brought about by improving the generalization ability.
> These works are aware of the difference between rain in the training set and real test images. But their solution is somehow to include real test images into the training set, even if we don't have corresponding clean images. These methods are effective when we can determine the characteristics of the test images. But this does not solve the generalization problem. Because these methods manage to convert "rain outside the training set" to "rain inside the training set". Since data collection is extremely difficult, these methods may still face challenges in practice.
>
> We would like to re-emphasize that the generalization problem in low-level vision mainly refers to model performance when test degradation is inconsistent with training degradation. Although many tasks can be improved through engineering methods (collecting many real images, simulating image degradation with more realistic means, etc.), these methods are not essentially trying to solve the generalization problem. These methods are still trying to overfit the real degradation. We can not deny the practical significance of these methods, but we want to emphasize the importance of generalization problems in low-level vision since there are a large number of tasks that cannot be solved without improving generalization performance. Understanding and improving generalization performance is an important research direction to solve these problems.
>
> To address the reviewers' concerns, we will include the above discussion in our next version of the paper. We will also include a comparison of these methods. At that time we can have further discussions on the new results.
>
> `Q5`: "Can authors also discuss how their study can be generalized to other low-level vision tasks?"
>
> `A5`: Thanks for your comment. The practical difficulties encountered by different low-level vision tasks may vary slightly, but the core problem is overfitting to training degradation. For different low-level vision tasks and different training data, the network overfits in different ways. In the words of this paper, the network slacks off differently. In the deraining task, the network is "slacking off" by memorizing the pattern of rain streaks. While in the super-resolution task, the network is "slacking off" by memorizing the frequency distribution of low-resolution images. The key insights our paper provides on the other low-level vision tasks are: (1) the generalization problem on these tasks is due to the overfitting to training degradation; (2) the existing inappropriate training objective is an important reason that the network overfitting; (3) learning and reconstruction of image distributions can help low-level vision models overcome generalization problems. We will integrate this discussion into our paper.
>
> `Additional Comment`:
>
> We notice that the reviewer thinks "Some of the paper’s claims have minor issues". We would like to ask if there are any issues other than those discussed above. If the reviewers can point it out, we can explain and discuss it further in the following discussion time. We hope to address any concerns of the reviewer.
>
> `Rebuttal References`:
>
> [R1] Semi-supervised transfer learning for image rain removal. CVPR2019.
>
> [R2] Syn2real transfer learning for image deraining using gaussian processes. CVPR2020.
>
> [R3] Memory-oriented transfer learning for semi-supervised image deraining. CVPR2021.

---

> ### Author Response · Authors · 2022-11-06
> **The First Response to Reviewer bcPF (Part 1)**
>
> Thanks to Reviewer bcPF for the comments. We'll update the experiments to address your concerns. We would like to discuss some of the issues you mentioned first so that we have more time to exchange views. We will update the paper when we have updated the results.
>
> `Q1`: "Observations from the experiments are not excellent.", "This conclusion statement is naive since this statement is very obvious and doesn't require any experiments and study to get to this conclusion.", and "conclusions and implications are naive"
>
> `A1`: We respectfully disagree with your comments.
>
> Firstly, our findings and observations are NOT naive. In this work, we argue that one of the reasons for the generalization problem of deraining models is that the model only focuses on learning rain patterns in the training set, rather than learning the content of natural images. More importantly, the reason for this phenomenon is that the network will slack off in the training settings that are commonly used today. The finding is groundbreaking because it goes against the common practice of using deep learning. In the existing deep learning practice, we strive to increase the diversity and complexity of training background images. However, our work finds that this tendency in deep network training is caused by the difference in the complexity of the network between the training background and the raindrop images during training. To this aspect, reviewers DKEu and LTSX also think our work "is in great conflict with existing understanding". Our findings point to the fact that existing research directions such as improving the network, increasing the diversity and complexity of training images, and synthesizing more realistic datasets may fail to address the root causes of generalization problems. This is the core value that our work brings to the low-level field. Because a lot of work today is still expected to make low-level models practical in real scenarios (overcoming generalization problems) simply by developing better network structures. Our work brings awareness to a long-overlooked problem in deep network training and underscores the significance of deep learning interpretability work.
>
> Our implications are also NOT naive. The knowledge demonstrated in our work not only provides direct insights into the deraining task, but also provides insights for improving the generalization ability of other low-level vision tasks. This is also recognized by reviewer LTSX. Although our conclusions sound simple, no existing work addresses similar conclusions nor does any work follow a similar idea of thinking and philosophy. Even, our conclusions are contrary to most existing work.
>
> It is possible that it follows the most straightforward logic from our findings to our method demonstrated in the implication. This may make our method seem too straightforward. We think this shows the value of our findings. Because such a simple change of concept can lead directly to a simple and effective method with potential. Along this line, more and better methods may be proposed in the future.
>
> So we respectfully disagree with your assessment of the "naive" of our work. We very much welcome the reviewers to provide some literature to illustrate the naivety of the content and results of this paper. We will explain to you further how our work differs from possible previous work during the discussion period.
>
> `Q2`: "It would have been better if authors propose a training strategy or a loss that addresses generalization problems in low-level visions."
>
> `A2`: Thanks for the comment. The purpose of this paper is to take the task of removing rain as a starting point to explore low-level vision generalization. Starting from our core findings, many possible paths can lead us to obtain a model with better generalization ability, such as designing new low-level pre-training models, designing loss functions to make the model focus on the content of images, or generating derained images through conditional generation. But this is all based on the new understanding brought about by this paper. We think this has demonstrated our contribution.
>
> In our work, we demonstrate the potential of direct implementation of our findings. We can further devise more complex and elaborate methods for better results, but this is beyond the scope of this paper and also beyond the limit of a 9-page paper. Our future work will apply our ideas to designing generalizable low-level visual methods. While we are always looking for better methods, we believe that interpretability work like ours is equally valuable and indispensable.

---

> ### Author Response · Authors · 2022-11-17
> **Follow Up Response to Reviewer bcPF**
>
> Dear reviewer bcPF. Wish you have a good day.
>
> We have updated the main text to include the discussion of semi-supervised methods and results of real-world rainy images. We cite and discuss [R1,R2,R3,R4] and compare with 2 Syn2Real models. One is Syn2Real-Syn trained only using the synthetic data, and another is Syn2Real-Real trained using labeled synthetic data and unlabeled real images. Due to the difference in the distribution of rain streaks, the models trained using synthetic data can not generate satisfactory rain removal effects. When obtaining some real images, Syn2Real-Real can indeed achieve some improvement. However, these improvements are not brought about by improving the generalization ability. Because these methods manage to convert “rain outside the training set” to “rain inside the training set”. Since data collection is extremely difficult, this method still faces great challenges in practice. Our method improves generalization performance and achieves better results on test images. We also include these discussions in the related work section in the appendix. We have also revised the conclusion section to include a discussion related to your question.
>
> Hope our response can solve your concerns. Please let us know if you have any unsolved or other concerns. Thanks.
>
> Best,
>
> References:
>
> [R1] Wei, W., Meng, D., Zhao, Q., Xu, Z., & Wu, Y. (2019). Semi-supervised transfer learning for image rain removal. In Proceedings of the IEEE/CVF conference on computer vision and pattern recognition (pp. 3877-3886).
>
> [R2] Yasarla, R., Sindagi, V. A., & Patel, V. M. (2020). Syn2real transfer learning for image deraining using gaussian processes. In Proceedings of the IEEE/CVF conference on computer vision and pattern recognition (pp. 2726-2736).
>
> [R3] Huang, H., Yu, A., & He, R. (2021). Memory oriented transfer learning for semi-supervised image deraining. In Proceedings of the IEEE/CVF Conference on Computer Vision and Pattern Recognition (pp. 7732-7741).
>
> [R4] Yasarla R, Sindagi VA, Patel VM. Semi-supervised image deraining using Gaussian processes. IEEE Transactions on Image Processing. 2021 Jul 16;30:6570-82.

---

> ### Author Response · Authors · 2022-11-22
> **Further discussion with Reviewer bcPF**
>
> Dear reviewer bcPF:
>
> We thank you for the precious review time and valuable comments. We have provided corresponding responses and results, which we believe have covered your concerns. We realized that we could no longer revise our manuscript. But we can continue the discussion to address your concerns.
>
> We hope to discuss further with you whether or not your concerns have been addressed. Please let us know if you still have any unclear parts of our work.
>
> Best,
>
> Paper 716 Authors.

---

### Author Response · Authors · 2022-11-17
**Response to all reviewers and area chairs for a brief summary**

We thank all reviewers for their precious review time and valuable comments. We would like to give a brief summary here.

What we have done during the rebuttal phase:

1. We **add the results on real-world rainy images** (reviewer bcPF and kgGu) in the revised paper. The results demonstrate the good generalization performance of the described simple method.
2. We compare the proposed findings and the described method **with semi-supervised deraining methods** (reviewer bcPF). We compare the Syn2Real method with different settings with ours on real-world test images. The results show the value and differences between the proposed findings and the previous efforts. We also include the discussion in the revised paper.
3. We re-explain the key insights of our work toward the low-level vision field (reviewer bcPF).
4. We add an experiment to study the model performance on background reconstruction and rain removal **when the training and testing image distributions are mismatched** (reviewer LTSX). The results show that our method is robust to training image content to a large extent.
5. We take some valuable suggestions proposed by reviewers, e.g., correcting the typos. We will keep revising this paper from the aspect of writing.

We would like to ask if there are any issues other than those discussed above? If the reviewers can point it out, we can explain and discuss it further in the following discussion time. We hope to address any concerns of all the reviewers.

Thanks for your effort in reviewing this paper,

Paper 716 Authors.

---

> ### Author Response · Authors · 2022-11-26
> **Call for discussion to all reviewers and area chairs - Deadline is Dec. 2**
>
> Dear Reviewers and ACs,
>
> We deeply appreciate your valuable comments and constructive feedback on our paper. We have responded to the reviewers' comments. We also revised the paper. We also summarized the specific changes we made.
>
> **We have sent several calls for discussion**. But, **NO reviewer responded to us**. While we were unable to revise the manuscript during the discussion further, we still hope to address possible concerns of the reviewers through the discussion. We believe that **participating in the discussion before the December 2 deadline** will help us better address reviewers' concerns. We would also like to ask the ACs to kindly remind the reviewers to participate in the discussion.
>
> Thanks for your effort in reviewing this paper,
>
> Paper 716 Authors.

---

### Decision · Program_Chairs · 2023-01-20

**Decision:**

Reject

**Justification For Why Not Higher Score:**

The conclusions of the paper are not supported by the experiments.

**Justification For Why Not Lower Score:**

N/A

**Metareview: Summary, Strengths And Weaknesses:**

The paper studies image deraining (removing rain from an image) with the goal of understanding the generalization problem in low-level vision tasks (i.e., image reconstruction tasks) more broadly.

The paper trains networks (res-net, U-net, and Swin-transformer) to reconstruct an image from an input image with rain corruptions. The paper argues that 'when too much background data is provided for training, the model cannot learn to reconstruct the image content and can only overfit to the degradation.' Thus reducing the number of background images can improve performance.
In the experiments, the paper investigates how well the networks perform in reconstructing the background (background reconstruction), i.e, the part that is not corrupted by rain, and in reconstructing the part corrupted by rain (rain removal performance).
The experiments demonstrate that decreasing the number of background images used for training can decrease the performance, a finding the authors describe as 'interesting and inspiring' and `groundbreaking'.

The paper makes relatively general statements from a small and narrow set of experiments. The paper in essence only conducts one experiment for deraining (and repeats that on four datasets), but draws very general conclusions, for example, it writes that `we argue that the generalization problem in low-level vision cannot be attributed to insufficient network capacity or training data.' The experiment provided in the paper doesn't give sufficient evidence for the broad statements made in the paper, and thus the paper is not ready for publication.


**Summary Of Ac-Reviewer Meeting:**

Here is a summary of the reviews:

R1 (5) notes that the paper is well written, but also notes that the conclusions are naive and that it's unclear how the results generalize to other low-level vision tasks. The authors disagree and describe their results as 'groundbreaking'. I agree with the general critique of the review that it is unclear what we can learn about general low-level vision tasks from the paper, and with the reviewer's statement that there is no evidence for the paper's claim that the relatively narrow experiments provide useful insight beyond image deraining.

R2 (6) notes that the finding that ``that training with fewer data and less complex background images can improve the generalization in deraining'' is an interesting and novel finding. The reviewer's main critique is that the paper 'doesn’t propose a better way to achieve the balance between restoring the image background and removing rain streaks', and notes that more experiments need to be done.
The authors respond that their main objective and contribution is to add to the understanding of low-level vision generalization, and not on how to improve it, or how to improve a better method.
I think it is fine to provide only an understanding on low-level vision generalization, without providing a better method, but the main issue is that the paper only provides a limited understanding on low-level vision generalization.

R3 (6) finds the finding interesting that 'existing derainig methods can be trained to have better generalization ability by reducing background images in training data'
The reviewer also notes that the finding is interesting, but the experiments are not sufficient to support the findings.

R4 (6) provided a relatively short review, stating that the author's perspective is interesting, and notes that a weakness is 'a lack of experiments on real-world rainy images' and that `there are many typos'. Both are not issues of the paper.

Based on my reading, the main issue of the paper is that it provides very broad conclusions that are not supported by the narrow set of experiments, as explained in the meta-review. The paper's approach is not sufficiently scientific in that it makes broad claims and assumptions without providing concrete evidence, examples are in the meta-review. An additional example is:
'In our study, we use the l1-norm loss to simplify the problem. In this setting, our conclusions do not lose generality and can be extended to other similarity-based training loss functions.'.
It is totally fine to use the l1 norm as a loss function, but there is no evidence that the conclusions apply beyond the specific setup of using the l1 norm as a loss function.